# Current Perspectives of Antifungal Therapy: A Special Focus on *Candida auris*

**DOI:** 10.3390/jof10060408

**Published:** 2024-06-06

**Authors:** Arumugam Ganeshkumar, Manickam Muthuselvam, Patricia Michelle Nagai de Lima, Rajendren Rajaram, Juliana Campos Junqueira

**Affiliations:** 1Department of Biosciences and Oral Diagnosis, Institute of Science and Technology, São Paulo State University (UNESP), São José dos Campos 12245-000, SP, Brazil; patricia.nagai@unesp.br; 2Department of Materials Physics, Saveetha School of Engineering, Saveetha Institute of Medical and Technical Sciences (SIMTS), Chennai 602105, Tamil Nadu, India; 3Department of Biotechnology, Bharathidasan University, Tiruchirappalli 620024, Tamil Nadu, India; muthuselvam@bdu.ac.in; 4Department of Marine Science, Bharathidasan University, Tiruchirappalli 620024, Tamil Nadu, India; drrajaram69@rediffmail.com

**Keywords:** *Candida auris*, antifungal therapy, antifungal natural products, antifungal peptides, antifungal essential oil

## Abstract

*Candida auris* is an emerging *Candida* sp. that has rapidly spread all over the world. The evidence regarding its origin and emerging resistance is still unclear. The severe infection caused by this species results in significant mortality and morbidity among the elderly and immunocompromised individuals. The development of drug resistance is the major factor associated with the therapeutic failure of existing antifungal agents. Previous studies have addressed the antifungal resistance profile and drug discovery for *C. auris*. However, complete coverage of this information in a single investigation is not yet available. In this review, we have mainly focused on recent developments in therapeutic strategies against *C. auris*. Based on the available information, several different approaches were discussed, including existing antifungal drugs, chemical compounds, essential oils, natural products, antifungal peptides, immunotherapy, antimicrobial photodynamic therapy, drug repurposing, and drug delivery systems. Among them, synthetic chemicals, natural products, and antifungal peptides are the prime contributors. However, a limited number of resources are available to prove the efficiency of these potential therapies in clinical usage. Therefore, we anticipate that the findings gathered in this review will encourage further in vivo studies and clinical trials.

## 1. Introduction

### 1.1. Candidiasis, Outbreak, and Epidemiology

Candidiasis is an infection caused by opportunistic pathogens of the *Candida* genus, with manifestations varying from mucocutaneous lesions to life threading bloodstream infections [1,2]. *Candida albicans* is the most common *Candida* species found in various human anatomical sites, including oropharyngeal, esophageal, gastrointestinal, and genital mucosa [3]. Other non-albicans species and *Candida*-related species are also simultaneously reported in the human body, such as *Nakaseomyces glabratus* (formerly *Candida glabrata*), *Pichia kudriavzevii* (formerly *Candida krusei*), *Candida parapsilosis*, *Candida tropicalis,* and others [1,4]. Recently, an emerging pathogen named *Candida auris* has been isolated from various clinical samples such as urine, stool, and vaginal, and rectal swabs. Similar to other *Candida* species, patients with comorbidities or a weakened immune system, or who have previously been exposed to antifungals or subjected to prolonged stays in healthcare settings are at risk for *C. auris* infection [5,6]. 

*C. auris* is recognized as an emerging fungal pathogen because of its widespread distribution, multidrug-resistant (MDR) behavior, high transmissibility, strong association with nosocomial infections and high mortality rates. *C. auris* expresses numerous virulence traits as well as tolerance to common antifungals, leading to therapeutic failure when treated with the most common classes of antifungals, including azoles, polyene, and echinocandins. In 2009, the first isolate of *C. auris* was reported from a Japanese female patient with ear discharge. In the same year, twelve isolates were obtained from otitis media patients in South Korea. Between 2009 and 2011, twelve isolates of *C. auris* were identified in India in patients with bloodstream infections. The first outbreak was noted during 2016 and 2017 in Europe and the United States of America (USA), respectively. Subsequently, *C. auris* has been reported in more than 47 countries according to the Centre for Disease Control (CDC) [7,8], and, nowadays, it is recognized as a threat to global society by the World Health Organization (WHO). Based on susceptibility profiles, outbreak potential, and clinical manifestations, *C. auris* isolates have been broadly classified into five clades (Figure 1). Recently, unpublished data indicated a sixth clade of *C. auris* isolated from samples collected in Singapore. The uncommon Clade VI isolates differed significantly from all other isolates in terms of their antifungal resistance genes, mating type locus, and chromosomal rearrangements. Although *C. auris* isolates have been widely investigated, the real rate of prevalence remains uncertain due to the lack of a proper dataset [9]. 

Furthermore, the origin of *C. auris* has not yet been elucidated and some authors speculate that global warming may be a possible reason for its spread [10]. *C. auris* cells remain viable for several months on environmental surfaces and medical equipment and prefer to colonize the skin of patients rather than other mucosal surfaces, leading to high probability of person-to-person transmission [11]. Since cells live freely on biotic and abiotic surfaces and tolerate antifungal and decontamination agents, the eradication of these fungal cells is extremely complicated [12,13,14,15]. All these factors make *C. auris* a global threat for immunocompromised patients in healthcare settings.

### 1.2. Drug Resistance: Molecular Bases

Clinical breakpoints of the existing antifungals for *C. auris* still need to be established. However, CDC has recommended the use of clinical breakpoints established from other closely related *Candida* species. The general guidelines defined to correlate the antifungal resistance of *C. auris* are ≥32, ≥2, ≥4, ≥2, and ≥4 for fluconazole, amphotericin B (AmB), anidulafungin, caspofungin, and micafungin, respectively [16]. Most clinical isolates of *C. auris* are multidrug-resistant or pan-resistant, especially for fluconazole, followed by AmB and voriconazole [17]. The wide administration of fluconazole for *Candida* prophylaxis and the use of azole derivatives in various applications, such as agriculture, might be the primary factors contributing to the development of azole resistance. Echinocandins are the only choice for *C. auris* infection treatment, although some studies already reported echinocandin-resistant strains [18]. 

The prevalence of drug-resistant *C. auris* strains has been analyzed in several studies worldwide. Isolates from India, Pakistan, South Africa, and Venezuela presented resistance rates of 93% to fluconazole, 35% to amphotericin B (AmB), and 7% to echinocandins, respectively. [6]. A resistance rate of 90% for fluconazole was observed in isolates from India and the USA, whereas all the isolates from the United Kingdom (UK) showed 100% resistance to fluconazole. AmB was the second most common antifungal drug related to *C. auris* resistance, with 20–30% of resistant isolates in India [19], 30% in USA [20], 62.5% in Saudi Arabia [21], 23.2% in Kuwait [22], and 33.3% in Oman [23]. 

Recent studies investigated the molecular bases of drug resistance in *C. auris* isolates. Different mechanisms of azole resistance have been discussed, including mutation in target genes (*ERG11*) [24], overexpression of genes encoding drug transporters, and an increased copy number of the TAC1B gene [25]. Mutation in the TAC1B gene was the second most observed mutation in *C. auris* after ERG11. Mutations like F126L, Y132F, and K143R were frequently noted in the ERG11 gene in fluconazole-resistant isolates, whereas A640V, A657V, and F862_N866del mutations were observed in the TAC1B gene in fluconazole-resistant isolates [25,26]. 

Mechanisms of resistance to echinocandins are mainly associated with the FKS1 gene involved in the biosynthesis of glucan and maintains the integrity of the cell wall. As occurs with *C. albicans* and *C. glabrata,* FKS1 is the target of echinocandins and mutations such as amino acid substitution and deletion in hot spots of FKS1 result in a lack of therapeutic effectiveness [27,28]. The most frequently observed FKS1 mutations are S639F, S639P, and S639Y in hot spot 1; however, less common mutations in hot spot 1 have also been reported (F635 del substitutions, F635L/Y, S639T, and D642Y) [29,30]. 

Regarding AmB resistance, a probable mechanism of action has not yet been elucidated. The increased Minimum Inhibitory Concentration (MIC) observed in AmB-resistant *C. auris* isolates were associated with mutation in the ERG6 gene [29]. Due to a loss of sterol methyltransferase activity, these isolates accumulate more cholesta-type sterols [26] Similarly, in vitro studies have evidenced that AmB-resistant strains carry nonsense mutations in genes like ERG11 and ERG3 [24].

## 2. Therapeutic Options 

### 2.1. Current Approved Antifungal/Considered Antifungals

The antifungal pipeline for *C. auris* infections includes various promising antifungal candidates, such as rezafungin, ibrexafungerp (IBX), fosmanogepix, and T-2307 (Figure 2). Rezafungin is a potential antifungal derived from anidulafungin that has exhibited strong in vitro antifungal activity against *C. auris*, including strains resistant to other echinocandins. It features a long half-life allowing for a once-weekly dosing schedule. In clinical trials, rezafungin has demonstrated a favorable safety profile with minimal side effects. On 31 March 2023, Rezzayo™ (Cidara Therapeutics, Inc., San Diego, CA, USA) was approved in the USA to treat adults with candidemia and invasive candidiasis. Nevertheless, it is still crucial to confirm its efficiency through data obtained from other clinical investigations. The high cost of this antifungal puts a barrier to its use in clinical practice in comparison with conventional treatments. 

Due to their notable superior antifungal activity, researchers have been focusing on IBX. This drug is a class of (1,3) β-D-glucan synthase inhibitors like echinocandins, a semi-synthetic derivate of enfumafungin. The mechanism of binding of IBX was not disturbed by the FKS1 gene mutation, which limits the cross resistance [32]. During preclinical investigations, its potential as a therapeutic agent for managing highly resistant *Candida* infections was confirmed [33,34]. IBX remains active against fluconazole and echinocandin-resistant *Candida* isolates and multidrug-resistant *C. auris*. However, IBX has shown a concentration-dependent fungicidal activity against various clinically relevant fungi, including *C. auris* [33,34,35,36].

Fosmanogepix, also known as APX001, is also a promising antifungal agent exhibiting the potential to treat *C. auris* infections. Fosmanogepix is a prodrug that is actively converted into manogepix, the active molecule. Manogepix directly inhibits the function of the Gwt1 (cell wall transfer protein 1) enzyme, resulting in pleiotropic effects on fungal cells, leading to severe growth defects and cell death. Clinical trials provided encouraging results regarding its efficacy and safety against this multidrug-resistant fungus. Fosmanogepix demonstrated strong activity against *C. auris* strains in laboratory tests, including those resistant to standard antifungals such as echinocandins. Phase II trials evaluating fosmanogepix against *C. auris* infections showed high treatment success rates. Apart from *C. auris*, fosmanogepix exhibited a broad spectrum of activity against other *Candida* species and filamentous fungus, making it a valuable tool for wider fungal infections. Clinical trials indicated that fosmanogepix was well-tolerated with minimal side effects. Serious adverse effects or treatment discontinuations were not reported. Fosmanogepix is available in both intravenous and oral forms, offering flexibility in treatment administration based on patient needs and disease severity [37]. Multiple Phase III clinical trials are currently investigating fosmanogepix for different fungal infections, including invasive candidiasis caused by *C. auris*. These trials will further evaluate its efficacy and safety on a larger patient population. Fosmanogepix is not yet approved for clinical use in any country. While the promising results from Phase II trials are encouraging, further research and evaluation through Phase III trials are still required [38].

Interestingly, a novel arylamidine (T-2307) expressed broad-spectrum antifungal activity against various fungal pathogens including several *Candida* species. Recent investigations highlighted the in vitro and in vivo efficiency of T-2307 on *C. auris*. Its mechanism of action was associated with the collapse of mitochondrial membrane potential by targeting respiratory chain enzymatic complexes III and IV of the yeast [39,40,41]. Phase I clinical trials evaluating the safety and tolerability of T-2307 in healthy volunteers were recently completed, showing promising results. Phase II clinical trials to assess its efficacy in patients with fungal infections are planned to begin in 2024. Considering its promising preclinical data and ongoing clinical trials, T-2307 could be approved for clinical use against *C. auris* infections within the next few years. 

### 2.2. Chemicals as an Emerging Weapon against C. auris 

Whilst some antifungal agents are in the clinical trials phase, the scientific community has worked intensely to discover other therapeutic options. Here, several new compounds with activity against *C. auris* are listed (Table 1) and discussed. Toepfer and collaborators investigated the compound clorgyline and its derivatives, which worked as multi-targeting inhibitors of the Cdr1 and Mdr1 efflux pumps of *C. albicans* and *C. glabrata.* Especially, the clorgyline analogs M19 and M25 expressed a high ability to inhibit the efflux pump activity of *C. auris* [42]. Recent investigations have discovered the activity of pyrazole moiety compounds against fluconazole-resistant *C. auris* isolates. These compounds exhibited broad spectrum, high potency, high selectivity, low cytotoxicity, and anti-drug resistance [43]. Novel tetrazoles featuring isoxazole moiety have also been identified as highly selective antifungal agents, displaying outstanding antifungal activity against fluconazole-resistant strains of *C. albicans*, *C. glabrata,* and *C. auris* [44]. A novel benzoanilide antifungal (compound A1) showed potent activity against *C. auris* cells through the blocking of virulence biosynthesis and altering of the cell wall of *C. auris* [45]. Furthermore, a compound named NSC319726 (thiosemicarbazone zinc chelator) exhibited MIC values ranging from 0.125 to 0.25 mg/L for *C. auris* isolates belonging to five different clades, expressing fungistatic activity in time-kill curves [46].

Many authors have focused on the study of Manogepix, a pyridine-isoxazole-based antifungal agent. Manogepix was active against most clinical isolates of *C. auris* belonging to South Africa. Over 300 *C. auris* isolates were studied, including 335 fluconazole-resistant, 19 fluconazole and AmB-resistant, 1 AmB-resistant and 2 pan-resistant. The MIC values of manogepix ranged from 0.002 to 0.063 μg/mL for fluconazole-resistant isolates, 0.004 to 0.031 μg/mL for fluconazole and AmB-resistant isolates, and 0.004 μg/mL to 0.008 μg/mL for pan-resistant isolates. The activity of manogepix was higher than azoles (3-fold), 4-fold higher than echinocandins, and 9-fold higher than AmB [50]. Manogepix also showed activity against clinical isolates from the New York outbreak, with MIC values of 0.008 to 0.015 mg/L against pan-resistant isolates [51]. A recent study has observed an increase in efficiency of anidulafungin against *C. auris* when combined with manogepix or 5-flucytosine [55]. 

Various studies have reported the biological role of metallic gold and its salts. Among them, gold(I)—phosphine complexes and the gold salt auranofin were tested against a panel of 28 fungal strains including *Candida* spp., *Cryptococcus* spp., *Aspergillus* spp., and *Fusarium* spp. Notably, two (complex 4 and 6) square-planar gold(I) complexes produced a remarkable antifungal activity in most of the tested isolates. Regarding *C. auris* isolates, complex 4 and 6 resulted in MIC values ranging from 3.9 to 7.8, and 1.95 µg/mL, respectively. However, auranofin did not produce considerable results (MIC > 31.3 µg/mL) against *Candida* species [52].

The activity of new promising antifungal compounds has also been investigated against *C. auris* in biofilm stage. Among several phenylthiazole small molecules, compound 1 emerged as the most potent antifungal, inhibiting the growth of *C. albicans* and *C. auris* strains at concentrations ranging from 0.25 to 2 µg/mL. This compound reduced 50% of metabolic activity of biofilm produced by *C. auris* with similar activity to AmB [53]. A class of molecule ceragenins was also found to inhibit both the planktonic and biofilm form of *C. auris*. Promisingly, they led to significant reductions in fungal infections in ex vivo mucosal tissues [54]. 

### 2.3. Essential Oils Are Potential Sources of Novel Antifungal Skeletons

Essential oils are mixtures of volatile compounds obtained through the steam distillation of various plant parts, such as flowers, bark, leaves, and other plant materials. Due to their unique properties, essential oils have a wide range of applications, with a predominant use in aromatherapy. Due to the presence of diversified volatile compounds, essential oils have been extensively studied as antimicrobial agents against bacteria and fungi, including *C. auris* strains.

For instance, Parker et al. demonstrated the effectiveness of selected essential oils against *C. auris* and found superior activity for the oils from cinnamon leaf, clove bud, lemongrass, and basil. The effective eradication of *C. auris* occurred with MIC values ranged from 0.01% to 1.0%. The same study reported the interactions between conventional antifungal drugs and essential oils. Clove bud oil synergistically interacted with fluconazole and flucytosine to combat *C. auris* [56]. 

Essential oil extracted from the seeds of *Withania somnifera* was also tested against *C. auris*, producing IC50 at 5.96 mg/mL and a fungistatic mechanism confirmed by killing assay. Its mechanism of action was associated with disturbing the membrane integrity of *C. auris* cells, evidenced through ergosterol binding and sorbitol protection assays. However, the seed oil was inactive against the mature biofilm formed by *C. auris* [57].

In a more detailed study, Di Vito et al. tested 15 essential oils against 10 clinical strains of *C. auris*. The results indicated that *Cinnamomum zeylanicum* essential oil was most effective against *C. auris* (MIC; 0.06% *v*/*v*) in synergy with the antifungal drug fluconazole. Further they verified that cinnamaldehyde was the sole reason for the antifungal activity [58]. In the same year, another group of researchers identified the antifungal potency of *Cinnamomum cassia* essential oil [59], this oil also being rich in chemical compounds like cinnamaldehyde [60]. The level of cinnamaldehyde in plants may vary depending upon the species; however, *C. cassia* and *C. zeylanicum* contain 85.3% and 90.5% cinnamaldehyde respectively [61]. 

### 2.4. Natural Products against Candida auris

The world of medicine is increasingly turning to nature’s bounty for solutions to modern health challenges. In the realm of fungal infections, a fascinating arsenal of weapons lies hidden within plants, microbes, and even marine organisms. These diverse and potent molecules, meticulously crafted by living organisms, offer a promising alternative to traditional antifungal drugs. With the rising tide of fungal resistance to existing therapies, natural products present a glimmer of hope in the fight against these tenacious pathogens. A list of natural products with anticandidal activity is presented in Table 2 and Figure 3.

Among the compounds from plants, great focus has been given to carvacrol, a phenolic monoterpenoid found in the essential oils of oregano, thyme, pepperwort, wild bergamot, and other plants. Due to its broad spectrum of biological responses [73]. Carvacrol was active against *C. auris* by modulating the expression level and action of certain antioxidant enzymes [69]. As with carvacrol, geraniol is another monoterpene alcohol found in geranium oil as a major component. Recently, Fatima and collaborators utilized geraniol against *C. auris*. Geraniol displayed fungicidal activity and an inhibition effect on the metabolically active biofilm of *C. auris*. Subsequently, geraniol improved the survival rate of *C. elegans* infected by *C. auris* [70]. 

Penta-O-galloyl-β-D-glucose (PGG), a bioactive product of many plants (firstly isolated from the leaves of *Schinus terebinthifolia*), was also investigated against *C. auris*. Chemically, PGG is a hydrolysable tannin reported to have plenty of biological activities such as antibacterial, anticancer, and antiviral activities. PGG demonstrated anticandidal activity within the MIC ranges of 1–8 µg/mL against drug-resistant *C. auris* [63]. Kim and Eom explored the antifungal and anti-biofilm properties of 6-shogaol against *C. auris*. Shogaols are pungent constituents of ginger and similar in chemical structure to gingerol [74]. The 6-shogaol demonstrated effectiveness in inhibiting the growth of *C. auris* in the concentration range of 16–32 µg/mL. Further, it showed promising activity in preventing the formation of biofilms and controlled the secreted aspartyl proteinase activity [72]. 

In relation to natural compounds from microbial sources, rubiginosin C obtained from the stromata of the ascomycetes *Hypoxylon rubiginosum* and *Hypoxylon texense*, effectively inhibited the formation of biofilms of *C. auris* and *C. albicans* [75]. In recent years, many studies have focused on the compound enfumafungin, a triterpene glycoside found in the culture supernatant of *Hormonema carpetanum*. This compound was used as a probe to produce biologically active antifungal IBX. Enfumafungin B and C, analogues of enfumafungin, were isolated. Both compounds were effective against clinically relevant *C. auris* with a MIC of 64 µg/mL. Further molecular docking studies confirmed that these compounds bonded in the transmembrane region of FKS1 of β-(1,3)-D-glucansynthase [64]. 

In another study, Persephacin, isolated from the endophytic fungus *Sphaceloma* sp., showed activity against a diverse range of fungal species (*C. albicans*, *C. glabrata*, *C. parapsilosis*, *C. krusei*, *C. kefyr*, *C. tropicalis,* and *C. auris*). Persephacin resulted in MIC values around 2.5 µg/mL, which was similar to the activity expressed by AmB [65]. A group of new linear lipopeptides (Myropeptin C–E and Myropeptin A1), isolated from the saprotrophic filamentous fungus *Myrothecium inundatum*, showed good inhibition of *C. auris* cells. In vitro hemolysis, cell viability, and ionophore assays indicated that these compounds target mitochondrial and cellular membranes, inducing cell depolarization and cell death [66]. Subsequently, hakuhybotrol along with six known cladobotric acids F, E, H, A, pyrenulic acid A, and F2928-1 were isolated from a culture broth of *Hypomyces pseudocorticiicola* FKA-73. Most of these compounds showed promising antifungal activity against azole-resistant and sensitive strains of *C. auris.* In particular, cladobotric acids F and E were able to cause a high inhibition in the fungal growth [67]. 

Although most antifungal compounds from microbial sources were isolated from fungus, some studies have investigated compounds from bacterial cultures. For example, a culture extract of *Lactobacillus paracasei* 28.4 inhibited several *C. auris* strains, acting against planktonic cells, biofilms, and persister cells. Further experiments confirmed that supplementation derived from *L. paracasei* 28.4 protected *G. mellonella* from *C. auris* infection [68]. 

Finally, compounds from marine sources have also been investigated against *C. auris*. Turbinmicin, a potent lead obtained from the marine reservoir (Turbinmicin-producing bacterium *Micromonospora* sp. WMMC-415.), displayed antifungal properties in several in vitro and in vivo experiments. Turbinmicin expressed a fungal-specific mechanism of action, targeting Sec14 of the vesicular trafficking pathway, a unique target, yet to be investigated [62]. A subsequent investigation showed that Turbinmicin exhibited a MIC value of 0.125 mg/mL and an inhibitory action on the mature biofilm of *C. auris* [76]. 

### 2.5. Peptide-Based Strategies for Eradicating C. auris 

Antimicrobial peptides (AMPs) are another alternative group of components frequently reported due to their superior biological properties. They are an active form of smaller segments of protein produced by various organisms, including plants, insects, humans, and other small animals. AMPs have distinct functions within the host; therefore, they express unexceptional behavior towards medically important pathogens like bacteria, fungi, and virus. On the other hand, the antimicrobial resistance mechanism of AMPs has not yet been fully elucidated. Wider investigations have identified some direct or indirect mechanisms of AMPs against pathogens, with a capacity to reduce virulence traits. As per the information available [77], there have been more than 3940 AMPs reported until now, including 3146 natural peptides and 190 predicted and 314 synthetic AMPs. More specifically, HsAFP1 (*Heuchera sanguinea*); NaD1 (*Nicotiana alata* flowers); Psd1 (*Pisum sativum* seeds); Psoriasin, CGA-N46, β-Defensin-1 to 4, and Histatin-5 from *Homo sapiens*; Gomesin, Heliomicin, Jelleine I to IV, and Lasioglossin I to III from insects and arachnids; and NFAP2 from filamentous fungi *Neosartorya fischeri* have been reported to have anticandidal activity [78]. A list of AMPs with their significant biological roles and their origins is presented in Table 3.

Among the microorganisms, both fungi and bacteria produce potent antimicrobial peptides. The fungus *Neosartorya fischeri* produces two distinct peptides: NFAP and NFAP2, with 57 and 52 amino acid lengths, respectively. The cystine residues present in these structures have remarkable importance, since they significantly improve the stability of the peptides at high temperatures [79,80]. NFAP and NFAP2 were identified as potent molecules against fluconazole (FLC)-resistant *C. albicans* [81] and *C. auris* [82]. Notoriously, NFAP2 interacted with most of the azoles and echinocandins, producing significant Fractional Inhibitory Concentration Index (FICI) values [82]. The bacteria *Bacillus subtilis* produces a lipopeptide (AF_4_) that showed a broad spectrum of antifungal activity on more than 110 fungal isolates. Recently, AF_4_ at 8 mg/L was found to kill most of *C. auris* cells. The mechanism of killing was associated with severe cellular membrane disruption and elevated generation of Reactive Oxygen Species (ROS) [83].

**Table 3 jof-10-00408-t003:** Antifungal peptides against *C. auris* derived from different classes of organisms.

S.no	Origin	Name	MIC Range	Reference
1	*Homo sapiens*	Human β-defensin-3	3.125 to 12.5 µg/mL	Shaban et al., 2023 [84]
2	*Homo sapiens*	Cathelicidin peptides LL-37	25–100 µg/mL	Rather et al., 2022 [85]
3	*Homo sapiens*	Histatin-5	7.5 µM	Pathirana et al., 2018 [86]
4	Barley plant	Defensin-like Protein 1 (D-lp1)	0.047–0.78 mg/mL	Kamli et al., 2022 [87]
5	*Bacillus subtilis*	AF_4_	8 µg/mL	Ramesh et al., 2023 [83]
6	American rattlesnake (*Crotalus durissus terrificus*)	Crotamine	40–80 µM	Dal Mas et al., 2019 [88]
7	*Pomacea poeyana*—freshwater snail	Pom-1, Pom-2	8.5, 8.4 µg/mL	Raber et al., 2021 [89]
8	Scorpion venom	ToAP1, ToAP2	>100 µM, 50–>100 µM	Pinheiro et al., 2023 [90]
9	Brilacidin	Semi synthetic	80 µg/mL	Dos Reis et al., 2023 [91]
10	Chemically prepared, Myristoylated and non-Myristoylated peptides	Pep-A, Myr-A, Pep-B, Myr-B, Pep-C, Myr-C	>256, >256, >256, 16–32, >256, 16–64	Bugli et al., 2022 [92]
11	Symbiotic NCR peptide fragments	NCR169C 17–38	6.25 µM	Szerencsés et al., 2021 [93]
		NCR169C 17–38 ox	12.5 µM	
12	Analogue of the peptide Cm-p5			
	Monomers	Cm-p5, Cyclic, Hcy	11 µg/mL, 27 µg/mL, Not active	Vicente et al., 2019 [94]
	Dimer	Dimer 1 (parallel)	30 µg/mL	
		Dimer 2 (anti-parallel)	31 µg/mL	
13	Rhesus macaque θ-defensin (RTD)	RTD–1, RTD–2	6.25, 6.25 µg/mL	Basso et al., 2018 [95]
	Olive baboon θ-defensins (BTD)	BTD–2, BTD–4, BTD–8	3.125, >25, 3.12–6.25 µg/mL	

The human body is also a significant source of antifungal peptides. So far, many peptides with different biological functions have been reported from the human body. Peptides like human β-defensin-3 [84], human cathelicidin peptides LL-37 [85], and salivary histatin-5 [86] were recently reported to have antifungal activity on *C. auris* (Table 3). Human β-defensin-3 and cathelicidin peptides LL-37 produced 100% and 70% of synergy with fluconazole [84,85]. The anticandidal activity of human histatin-5 was also documented against other non-albicans species, including *C. glabrata*, *C. parapsilosis*, *C. krusei*, *C. guilliermondii,* and *C. tropicalis* [96]. In addition, some studies highlighted the ex vivo effects of histatin-5 on mouse models, by reducing the fungal burden in both oral and vaginal infection models [97,98]. 

Plant defensins are potent molecules that protect plants from infections by effectively combating microbes without harming host cells. Certain plant-derived peptides, like HsAFP1 [99], NaD1 [100], Psd1 [101], and D-lp1 [102] exhibited strong activity against *C. auris*. Among them, D-lp1completely inhibited the biofilm formation and virulence of *C. auris* [102]. Similar to plant AMPs, other living organisms were found to be a source of antifungal molecules. Crotamine (from a South American rattlesnake) acted on multidrug-resistant *C. auris* with no cytotoxicity [88]. Pom-1 and Pom-2 peptides from *Pomacea poeyana* (the Cuban freshwater snail) inhibited *Pseudomonas aeruginosa* and fungi, including *C. auris* [89,102]. Scorpion venom peptides ToAP1 and ToAP2 showed promising antifungal effects against *C. auris*, alone and associated with other antimicrobial drugs [90] (Table 3).

### 2.6. Antifungal Immune Therapy against C. auris

Apart from AMPs, there are some other proteinous molecules that arouse interest as therapeutic strategies towards *C. auris* due to their immunological properties. These groups of molecules can be derived from the immune system of humans or certain animals. Amongst them, the complement receptor 3-related protein (CR3-RP) is one of the key surface antigens expressed during the biofilm formation of *Candida* species. Previous investigation identified the presence of CR3-RP moieties on the surface of *C. auris*. Upon in vitro exposure to prepared anti-CR3-RP, *C. auris* cells failed to form biofilm, confirming the ability of anti-CR3-RP for eradicating *C. auris* biofilms [103]. Similarly, Singh et al. utilized the anti-Hyr1p monoclonal antibody (mAb) to control the *C. auris* infection. The anti-Hyr1p mAb prevented biofilm formation and enhanced opsonophagocytic killing of *C. auris* by macrophages. In vivo studies showed that anti-Hyr1p mAb protected 55% of mice from the systemic infection caused by *C. auris* [104]. Other than these, NDV-3A (a vaccine based on the N-terminus of the Als3 protein formulated with alum) also showed effects against *C. auris*, inhibiting biofilm formation and encouraging the macrophage-mediated killing of *C. auris* [105]. 

A new humanized antibody H5K1 was recently identified and found to be active against *C. auris*. H5K1 expressed significant results when tested alone or in combination with Caspofungin and AmB [106]. Recent findings suggested that H5K1 specifically binds to β-1,3-glucans derived from *C. auris*, causing perturbation and remodeling of the fungal cell wall and facilitating the loss of cellular membrane integrity [107]. In support of this investigation, other *Candida* cell surface-specific mAbs were investigated in a mouse model of a *C. auris* invasive infection. For example, the specific monoclonal antibody C3.1, which targets the β-1,2-mannotriose (β-Man3) of *C. auris,* was able to improve the survival of animals and reduce the fungal burden in vital organs. In the same study, other peptide-specific mAbs such as 6H1 and 9F2 were reported as targeting hyphal-specific protein 1 (Hwp1) and phosphoglycerate kinase 1 (Pgk1), respectively. It also showed the same outcome as C3.1 in comparison with the control group. Altogether, the 6H1 + 9F2 cocktail enhanced the therapeutic outcome compared to the monotherapy. Therefore, all the three antibodies reported here might be an alternative to treat *C. auris* infections [108]. 

Intravenous immunoglobulins (IVIG) have been considered as an alternative therapeutic strategy to treat patients who present primary antibody deficiencies. They are the therapeutic product of normal human IgG [109,110,111]. Xin et al. recently demonstrated the role of IVIG in prevention and control of *C. auris* and *C. albicans* invasive candidiasis in animal models. Treatment with IVIG prolonged the survival and reduced the fungal burden in organs of the treated animals. In combination, IVIG enhanced the therapeutic index of AmB in comparison with monotherapy [112]. 

Immuno-informatics-based approaches offer an alternative and burgeoning avenue for designing suitable vaccine candidates against fungal infections. Recently, subtractive proteomics approaches were employed to design vaccines against *C. auris*. Khan et al. adopted this method and generated multi-epitope vaccine candidates, which elicited immune responses against *C. auris* infection by inducing various immune factors such as IgM, IgG, IL-6, and Interferon-α [113]. Similarly, Gupta and colleagues developed a vaccine based on novel CD4+ epitopes through genome-wide scanning and a reverse vaccinology approach. This vaccine has a reduced chance of becoming ineffective because the rapidly evolving genes of *C. auris* were eliminated from the epitope selection process [114].

### 2.7. Antimicrobial Photodynamic Therapy (APDT) against C. auris

APDT is a promising approach as an adjuvant therapy for fungal infections. As an advantage, APDT simultaneously targets different biomolecules of pathogens. Thus, it does not have any specific mechanism of action, which restricts the development of cross resistance. Therefore, APDT is considered an alternative therapeutic strategy against multiple clinically important organisms, including *C. auris*. APDT involves the association of a photosensitizer with irradiation from a light source, resulting in the generation of ROS, a prime factor responsible for the effectiveness of APDT. 

Phenothiazinium-based photosensitizers are promising agents proven to deactivate *C. auris* cells. Methylene blue, toluidine blue, new methylene blue, and the pentacyclic derivative S137 were assessed as photosensitizers for APDT on *C. auris* (CDC B11903). Their efficacy was evaluated by the determination of MIC and the use of a *G. mellonella* insect model. Based on the findings, the pentacyclic derivative S137 was identified as a potent treatment for *C. auris* [115]. Previously, it had demonstrated strong inhibition of *C. albicans* [116]. In another study, researchers utilized red, green, and blue visible lights alone and in combination with photosensitizers (new methylene blue, toluidine blue O, and rose bengal) against *C. auris*. The results showed that blue light alone disturbed the mature biofilm, but it was significantly improved when the photosensitizer was combined. On the other hand, red or green light alone had no effect on the *Candida* biofilm. The biofilms were disturbed only with the combination of light and photosensitizers [117].

Recently, Silva and their coworkers assessed the impact of methylene blue and 1,9-dimethyl methylene blue in addition with a red light-emitting diode (LED) on *C. auris*. At 3 μM, regardless of the light dose, 1,9-dimethyl methylene blue reduced the metabolic activity of *Candida* cells. Furthermore, it promoted high levels of ROS, lipid peroxidation, and mitochondrial membrane damage. In contrast, methylene blue was active only at a concentration of 100 μM when exposed to the highest dose of light. Further, studies evidenced that 1,9-dimethyl methylene blue was capable of inhibiting biofilm formation and the mature biofilm formed by *C. auris* [118]. Earlier studies by Stefanek and his team confirmed the positive effect of methylene blue with a red laser on the biofilm of *C. auris*. Up to 90% biofilm inhibition after 300 s of irradiation was observed compared to the growth control. Interestingly, in the presence of 0.25 mM methylene blue, the expression of both MDR1 and CDR1 genes was affected [119].

### 2.8. Repurposing of Drugs with Antifungal Properties

Drug repositioning or repurposing is a process of utilizing commercially available drugs for treating diseases outside the scope of its original indication. Drug repurposing is considered an important approach to manage emerging diseases caused by bacteria, fungi, and viruses [120,121]. The availability of various information rather than therapeutic indexes is a valuable point to consider in the repurposing process. Drug repurposing can reduce the time and cost of new drug development since it leverages existing data on toxicity profiles and preclinical parameters [122]. Therefore, the repurposing of drugs is recognized as an alternative to combat antifungal drug resistance, and several commercial non-antifungal drugs with activity against *C. auris* have been reported (Figure 4). Among them, Sertraline (an antidepressant agent that comes under serotonin reuptake inhibitors) showed activity against three different isolates of *C. auris*, possessing efficient antifungal activity by suppressing the action of yeast to hyphae conversion and biofilm formation [123].

Lohse and their colleague have aimed to develop antifungal agents from a group of FDA-approved compounds [124]. The selection of these compounds was based on MIC values of <10 µM. Among the hydroxyquinolines tested, clioquinol was more active than others; however, the authors were unable to investigate their mechanisms of action [124]. In another study, a hydroxyquinoline derivative known as nitroxoline was also active against 35 isolates of *C. auris* with MIC values ranging from 0.125 to 1 μg/mL. The resultant MIC values were lesser than the activity of fluconazole and AmB. Finally, nitroxoline was recommended for the treatment of *C. auris*-mediated candiduria. However, in vivo and clinical efficiency remains uncertain [49].

Synergistic drug interactions have also been investigated to increase the success of drug repurposing [125]. Pitavastatin, a cholesterol-lowering drug, proved to be a potent azole chemosensitizer. It reduced *Candida* biofilm formation and lowered MIC ranges when combined with fluconazole against *C. auris* [126]. Aprepitant, an antiemetic drug, showed the ability to disrupt metal ion homeostasis in *C. auris*, synergizing with azoles to reduce MIC by up to eight-fold and inhibiting biofilm formation by 95 ± 0.13% [127]. Miltefosine, an antiparasitic drug licensed for leishmaniasis, demonstrated potential against *C. auris* and other *Candida* strains, especially in combination with other antifungal drugs [128,129]. Colistin, an antibiotic used against multidrug-resistant Gram-negative infections, such as pneumonia, showed synergistic effects when combined with caspofungin, with FICI values ranging from 0.08 to 0.14. However, combining colistin with micafungin showed indifferent results, with FICI values ranging from 0.51 to 1.01 [130]. Synergistic combinations between HIV protease inhibitors and azoles were also found to be active against drug-resistant *C. auris*. Lopinavir combined with itraconazole achieved potent effects, increasing the survival rate of *C. auris*-infected *C. elegans* by up to 90% and reducing fungal burden by 88.5% [131]. Additionally, lopinavir and ritonavir interacted synergistically with itraconazole, effectively combating disseminated candidiasis in a rat model [132]. Atazanavir resensitizes *C. auris* to azoles by inhibiting efflux pumps, glucose transport, and ATP synthesis [133]. Moreover, the combination of saquinavir and itraconazole significantly reduced fungal burden in murine models, with an 88% decrease in colony-forming units compared to itraconazole alone [134].

Recently, some studies validated the synergistic potential of azoles in combination with chlorhexidine, used as a skin antiseptic and mouthwash due to its broad-spectrum antibacterial effects. It was reported that chlorhexidine can bind to cellular membrane phospholipids, causing changes in osmotic pressure and promoting cell lysis [135]. When combined with fluconazole, chlorhexidine significantly reduced the viability of both planktonic and biofilm forms of *C. auris* [136]. These results suggest the combined use of chlorhexidine and azoles to control *C. auris* infections in cutaneous and mucosal surfaces. Sulfamethoxazole is another class of antibiotic usually used to treat a variety of infections of the urinary tract, respiratory system, and gastrointestinal tract. Earlier investigation identified synergist interaction of sulfamethoxazole with azole antifungal drugs against *C. albicans*. The present study highlighted that sulfamethoxazole in combination with voriconazole and itraconazole restored the fungistatic potency of both drugs. Further, the superiority of the azole–sulfamethoxazole combination against *C. auris* infection was proved in *C. elegans* [137].

### 2.9. Nanotechnology-Mediated Antifungal Therapy

Metallic nanoparticles have been investigated as antimicrobial agents against a large number of microorganisms. The detailed investigations of the potent antifungal properties of different nanoparticles are presented in this review. In the last decades, researchers have extensively worked to develop different metallic nanoparticles, such as Ag, Zn, and Au, targeted to combat pathogenic microorganisms.

Regarding *C. auris*, most studies have focused on silver-based nanoparticles. Humberto and colleagues verified that silver nanoparticles effectively limited biofilm development at a silver nanoparticle concentration of 0.48 ppm, suggesting their use for controlling *C. auris* in healthcare settings [138]. Another study investigated silver nanoparticles for effectiveness against multidrug-resistant *C. auris*, showing strong antifungal properties with a MIC of <0.5 μg/mL on planktonic cells and a Minimum Biofilm Inhibitory Concentration (MBIC) of <2 μg/mL on preformed biofilm [139]. Consistent findings also revealed a significant reduction in viable *C. auris* cells, in both planktonic and biofilm form, upon treatment with silver nanoparticles [140].

More recently, several functionalized silver nanoparticles have been produced by green synthesis using plant compounds as metal ion-reducing agents. Polyphenol-capped metallic silver nanoparticles, such as those derived from *Cynara cardunculus* extract, exhibited an antifungal effect on *C. auris* by inducing mitochondrial toxicity and DNA fragmentation at a concentration of 50 µg/mL [141]. Trimetallic (Ag-Cu-Co) nanoparticles, synthesized with compounds from *Salvia officinalis*, also showed potent antifungal properties, inducing apoptosis and G2/M phase cell cycle arrest in *C. auris*, with MIC values ranging from 0.39 to 0.78 μg/mL and Minimum Fungicidal Concentrations (MFC) ranging from 0.78 to 1.56 μg/mL [142].

In addition to silver nanoparticles, various other metal nanoparticles have demonstrated activity against *C. auris*. For example, bismuth nanoparticles have shown promising effects in combating multidrug-resistant *C. auris*, exhibiting anticandidal activity with MIC ranging from 1 to 4 µg/mL, and disrupting both cells and biofilms of *C. auris* [143]. Caspofungin-loaded zinc oxide nanoparticles have demonstrated antifungal activity against caspofungin-resistant *C. auris.* Interestingly, caspofungin–ZnO nanoparticles did not induce acquired or cross resistance in *C. auris* [144].

### 2.10. Liposomal Technology for Efficient Drug Delivery

Liposomal technology represents a promising avenue for antifungal therapy, using various approaches to prepare liposomal delivery systems. The key point among these methods is the careful selection of lipid moieties for encapsulating a specific drug. This selection influences the surface charge of the liposome, enabling tailored delivery of potent molecules. The advantages of liposomal technology are manifold, including improved bioavailability, reduced toxicity, and targeted delivery. For instance, liposomal formulations of amphotericin B offer enhanced solubility and reduced nephrotoxicity compared to conventional amphotericin B, confirming the potential of liposomal technology in optimizing therapeutic outcomes.

De Alteriis et al. examined the role of liposomal technology to improve the antifungal effects of essential oil from *Lavandula angustifolia*, achieving antibiofilm activity with the persister-derived biofilm of *C. auris* [145]. Similarly, *Lippia sidoides* essential oil was loaded in lipid carriers, and the reduction in the MIC values were observed on *C. auris* [146]. In line with earlier investigations, Jaromin et al. utilized liposomal formulation to improve and modulate the surface characteristics of PQA-Az-13, which is the combination of indazole, pyrrolidine, and arylpiperazine scaffolds substituted with a trifluoromethyl moiety. Here, liposomes displayed a mean size of 76.4 nm, a positive charge of +45.0 mV, excellent stability, and no toxicity to normal human dermal fibroblasts. PQA-Az-13 showed MIC values between 0.67 and 1.25 µg/mL against *C. auris* and demonstrated promising results against in vitro biofilms and ex vivo skin colonization models [147].

## 3. Conclusions and Future Perspectives

The world is facing a crisis with the growing incidence of several infectious diseases and the limited availability of antifungal therapies, highlighting the urgent need to combat fungal infections. Compared to bacterial infections, fungal infections are less common, which has led to a limited range of antifungal drugs. However, targeting fungal-specific pathways plays a crucial role in the development of novel antifungal drugs. This limited choice of antifungal drugs contributes to their overuse, misuse, abuse, or prolonged use, leading to the development of antifungal drug resistance. The failure of antifungal therapy is associated with multiple factors, among which the development of drug resistance is the prime contributor. The identification of fungal-specific drug targets is a key characteristic of developing new antifungal therapies. This is crucial because they can effectively combat fungal infections without negatively affecting the host.

To address these challenges, there is an urgent need for innovative strategies in antifungal drug development. Especially, the elucidation of new drug targets, identification of novel drug scaffolds, designing of new drugs, and development of combined therapies are the most important points to consider. Additionally, some researchers are focusing on drug repurposing strategies. Adapting drugs already approved for other conditions can be a faster and cheaper way to bring new antifungals to market, especially against emerging or resistant fungal threats. While still in early stages, research on fungal vaccines is ongoing, focusing on stimulating the immune system to recognize and fight fungal infections. Tailoring antifungal treatment based on individual patient characteristics and the specific fungal strain involved in the infection can improve outcomes and reduce the risk of resistance.

Implementation of antimicrobial stewardship programs is crucial to prevent the overuse of common antifungal agents in medicine and other fields such as agriculture. By addressing these issues, there is a possibility to improve the management of severe fungal infections and reduce their impact on global health. Although several approaches discussed here showed promising antifungal activity against *C. auris*, few researchers extended their results to animal models and clinical trials. Overall, it is hoped that the data gathered in this review can provide support and insights into the advances of new treatments for *C. auris* infections.

## Figures and Tables

**Figure 1 jof-10-00408-f001:**
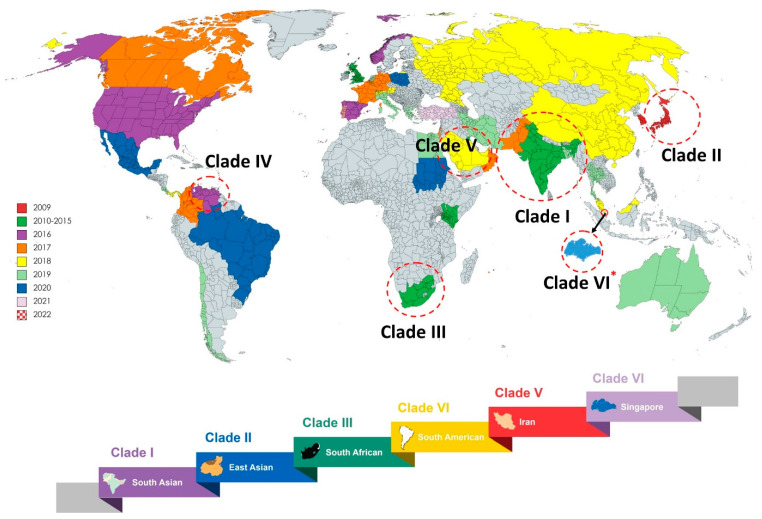
An updated illustration of distribution pattern of *C. auris* in various geographical locations around the world. The data presented here are based on the information obtained from recent publications. A color scheme was used to represent the year of *C. auris* reports. Over 40 countries reported positive for *C. auris* between 2009 and 2022. By 2022, five clades had been reported worldwide; however, recently unpublished data reported the existence of a sixth clade, suspected to have originated in Singapore [9] (* represents the preprint of the publication).

**Figure 2 jof-10-00408-f002:**
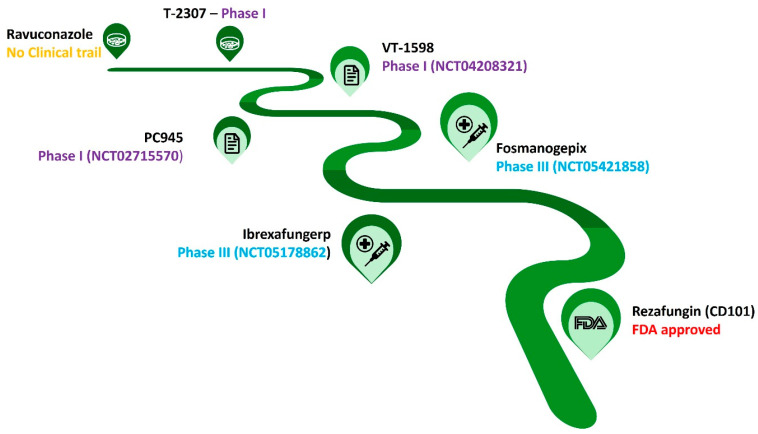
Currently available treatment options for *C. auris* prophylaxis. As of December 2023, only one molecule was approved by US-FDA to manage *C. auris*. Rezafungin is a modified version of anidulafungin, with the structural modification aimed to reduce the hepatotoxicity of the molecule while its efficiency was retained. The data presented here are based on the recent information provided by Wang et al., 2024 [31].

**Figure 3 jof-10-00408-f003:**
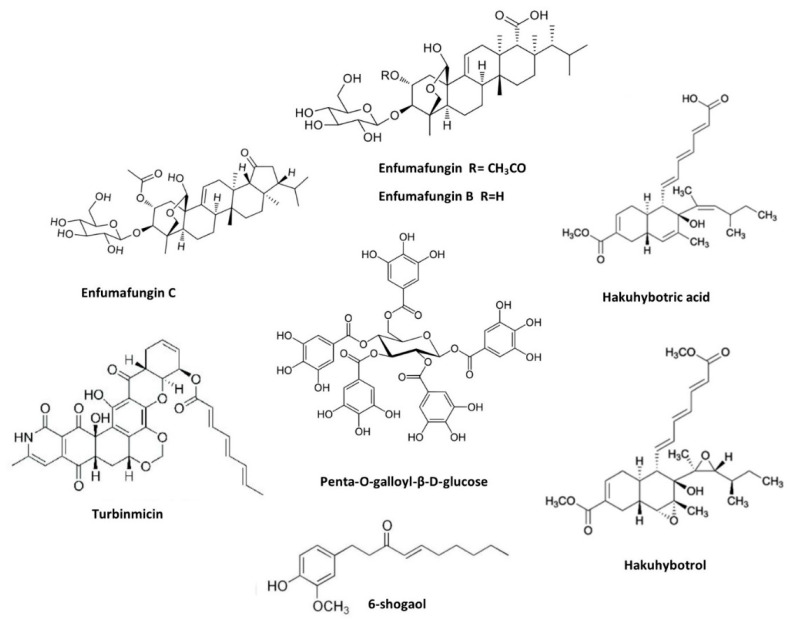
Scheme of potential molecules from different sources of natural products against *C. auris*.

**Figure 4 jof-10-00408-f004:**
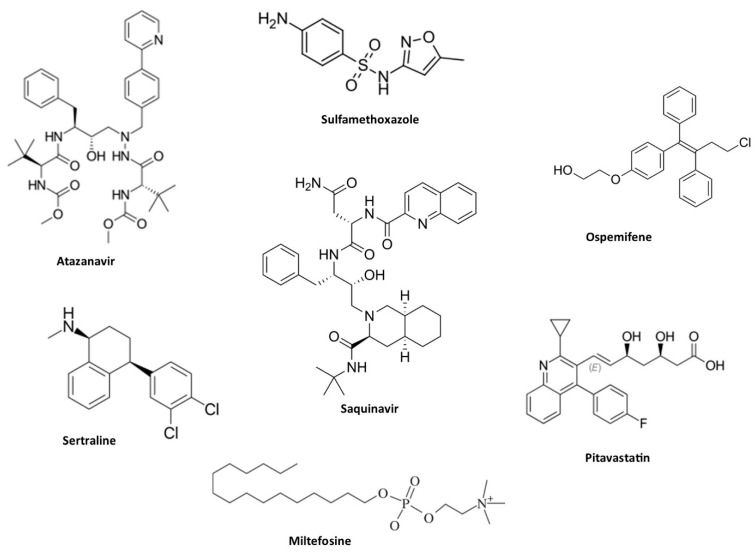
List of commercial drug molecules used to manage other disease conditions showing antifungal action against *C. auris*.

**Table 1 jof-10-00408-t001:** Minimal Inhibitory Concentration (MIC) of different classes of chemically derived compounds against *C. auris*.

Group	Compound	MIC or MIC Range	Reference
Clorgyline analogs	M19	86.7–98.3 µM	Toepfer et al., 2023 [42]
	M25	137–228 µM	
Novel tetrazoles featuring a pyrazole moiety	8, 11, 15, 24, 25	<0.0625–4, 0.125–8, <0.0625–2, 0.5–64, <0.0625–16 µg/mL	Chi et al., 2023 [43]
Novel tetrazoles featuring an isoxazole moiety	10d, 10h, 13r, 13u	0.008, <0.008, 0.0313, 0.0313 µg/mL	Ni et al., 2023 [44]
Benzoanilide group	Compound A1	0.5–2.0 µg/mL	Tu et al., 2023 [45]
Piperidine based 1,2,3-triazolylacetamide derivatives	pta1, pta2, pta3, pta4, pta5, pta6	0.48–0.97, 0.24–0.48, 0.12–0.24, >250, >250, >250 µg/mL	Srivastava et al., 2020 [47]
Pyrrolidine-based 1,2,3-triazole	P1–P10	0.97–62.5 µg/mL	Wani et al., 2023 [48]
Nitroxoline		0.125 to 1 µg/mL	Fuchs et al., 2021 [49]
NSC319726, a thiosemicarbazone		0.125 to 0.25 µg/mL	Li et al., 2021 [46]
Manogepix	Flu^R^	0.002 to 0.063 µg/mL	Maphanga et al., 2022 [50]
	Flu and AmB^R^	0.004 to 0.031 µg/mL	
	Pan^R^	0.004 and 0.008 µg/mL	
Manogepix	Pan^R^—New York	0.008 to 0.015 µg/mL	Zhu et al., 2020 [51]
Gold(I)—Phosphine Complex 4		3.9 to 7.8 µg/mL	Dennis et al., 2019 [52]
Gold(I)—Phosphine Complex 6		1.95 µg/mL	
Phenylthiazole—Compound 1		0.25–2 µg/mL	Mohammad et al., 2019 [53]
Ceragenins	CSA-44, CSA-131, CSA-142, CSA-144	0.5–1, 0.5–1, 2 to 8, 0.5–2 µg/mL	Hashemi et al., 2018 [54]

Flu^R^—Fluconazole-resistant; AmB^R^—Amphotericin B-resistant; Pan^R^—Resistant to more than two antifungals.

**Table 2 jof-10-00408-t002:** Antifungal natural products against *C. auris*.

Source	Molecule	MIC/MIC Range	Reference
Turbinmicin-producing bacterium *Micromonospora* sp. WMMC-415	Turbinmicin	0.125–0.50 µg/mL	Zhang et al., 2020 [62]
*Hypoxylon rubiginosum* and *Hypoxylon texense*	Penta-O-galloyl-β-D-Glucose	1–8 µg/mL	Marquez et al., 2023 [63]
*Hormonema carpetanum*	Enfumafungin	64 µg/mL	Cheng et al., 2023 [64]
	Enfumafungin B	>64 µg/mL	
	Enfumafungin C	>64 µg/mL	
*Sphaceloma* sp. from the leaf of *Poplar* sp.	Persephacin	2.5 µg/mL	Du et al., 2023 [65]
*Myrothecium inundatum*	Myropeptin C	16 µg/mL	Jagels et al., 2023 [66]
	Myropeptin D	16 µg/mL	
	Myropeptin E	16 µg/mL	
	Myropeptin A1	4 µg/mL	
*Hypomyces pseudocorticiicola* FKA-73	Hakuhybotrol	>128 µg/mL	Watanabe et al., 2023 [67]
	Cladobotric acids F	>128 µg/mL	
	Pyrenulic acid A	16 µg/mL	
	F2928-1	2 µg/mL	
	Cladobotric acids E	2 to 4 µg/mL	
	Cladobotric acids H	16 to 32 µg/mL	
	Cladobotric acids A	4 to 8 µg/mL	
*Lactobacillus paracasei* 28.4	Culture extract	3.75 to 7.5 mg/mL	Rossoni et al., 2020 [68]
Oregano or thyme and other aromatic plants.	Carvacrol	125–500 µg/mL	Ismail et al., 2022 [69]
Medicinal plants	Geraniol	225 µg/mL	Fatima et al., 2023 [70]
Different *Streptomyces* spp.	Nikkomycin Z	0.125 to >64 µg/mL	Bentz et al., 2021 [71]
Medicinal plants	6-shogaol	16–32 µg/mL	Kim & Eom, 2021 [72]

## Data Availability

Data is contained within the article.

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
