# Peer review of "Current Perspectives of Antifungal Therapy: A Special Focus on Candida auris"

_jof, 2024, doi:10.3390/jof10060408_

Round 1

Reviewer 1 Report

Good, interesting review addressing an important issue. Worth publishing in JoF after a minor revision is made.

line 120. 2.1. Current approved/considered antifungal therapy (change for: Current approved/considered antifungals)

Information on mechanisms of action/targets for Fosmanogepix and T-2307 should be provided.

Lines 166-167. Mechanism of action of T-2307 is not connected with inhibition of fungal cell wall biosynthesis. This agent induces collapse of fungal mitochiondrial membrane potential (ref. 35). Correct.

Lines 186-188. Should be changed for: .....blocking the biosynthesis of virulence factors and fungal cell walls through the inhibition of glycosylphosphatidylinositol (GPI) and GPI-anchored proteins.

Lines 197-205. This passage should be moved to section 2.8. Repurposing of drugs with antifungal properties.

Line 197. What the authors mean by "antifungal metabolites" in this sentence? This expression does not seem to fit here.

Fig. 4. Presence of sulfamethoxazole structure in this figure is surprising since there is no information on using this known antibacterial drug as an antifungal agent in section 2.8. Either remove this structure from the Fig.4 or provide an appropriate information in section 2.8.

Reference list

A number of citations lack the page range data. Should be complemented.

Author Response

  1. line 120. 2.1. Current approved/considered antifungal therapy (change for: Current approved/considered antifungals)

         Answer:  The manuscript revised as per the reviewer suggestion.      

  1. Information on mechanisms of action/targets for Fosmanogepix and T-2307 should be provided.

         Answer:  The mechanism of action of Fosmanogepix and T-2307 was                     provided.

  • Fosmanogepix is a prodrug that is actively converted into manogepix, the active molecule. Manogepix directly inhibits the function of the Gwt1 (cell wall transfer protein enzyme, resulting in pleiotropic effects on fungal cells, leading to severe growth defects and cell death.
  • T-2307 mechanism of action was associated to the collapse of mitochondrial membrane potential by targeting respiratory chain enzymatic complexes III and IV of the yeast.
  1. Lines 166-167. Mechanism of action of T-2307 is not connected with inhibition of fungal cell wall biosynthesis. This agent induces collapse of fungal mitochiondrial membrane potential (ref. 35). Correct.

         Answer: The mechanism of action of T-2307 was modified and                               appropriately presented. 

  1. Lines 186-188. Should be changed for: .....blocking the biosynthesis of virulence factors and fungal cell walls through the inhibition of glycosylphosphatidylinositol (GPI) and GPI-anchored proteins.

         Answer: An appropriate modification included.

  1. Lines 197-205. This passage should be moved to section 2.8. Repurposing of drugs with antifungal properties.

        Answer: As suggested by the reviewer section 2.8 was placed in an                        appropriate position.

  1. Line 197. What the authors mean by "antifungal metabolites" in this sentence? This expression does not seem to fit here.

         Answer: We are very much grateful to the reviewer for pointed out the                 errors in the manuscript. As suggested by the review all the errors in the               manuscript are addressed and highlighted for your reference.  

  1. Presence of sulfamethoxazole structure in this figure is surprising since there is no information on using this known antibacterial drug as an antifungal agent in section 2.8. Either remove this structure from the Fig.4 or provide an appropriate information in section 2.8.

        Answer: As suggested by reviewer an appropriate information is included in the revised manuscript.

Sulfamethoxazole is another class of antibiotic usually treatment of choice for variety of infections of the urinary tract, respiratory system, and gastrointestinal tract. Earlier investigation identified synergist interaction of with azole antifungal drugs against C. albicans. But this time it was extended up to C. auris. This study highlighted that sulfamethoxazole in combination with voriconazole and itraconazole restore the fungistatic potency of both the drugs. Further, the superiority of azole- sulfamethoxazole combination proved in vivo in a C. elegans model of C. auris infection (Eldesouky et al., 2018).

Reviewer 2 Report

The Review covers the different approaches previously published for the treatment of Candida auris. They include existing antifungal drugs, chemical compounds, essential oils, natural products, antifungal peptides, immunotherapy, antimicrobial photodynamic therapy, drug repurposing and drug delivery systems. 

However major revision is required. They are detailed below 

The manuscript is comprehensibly written

However, some major and minor corrections are required:

Major concerns

Line 233: In this phrase there is a serious conceptual error: You say: “Essential oils are the fatty acids moieties derived from bioactive plants.” Please consult to a Natural products Chemist. Essential oils are mixtures of volatile compounds that are the contrary of fatty acids, which are fixed oils. This is an error that cannot be published.

Lines 332-335: You say “These chemical substances alone or in combination with other drugs produce significant biological activities among the different targets.” Note that essential oils are not “chemical substances” but mixtures of volatile compounds, obtained by steam distillation

Minor remarks

Lines 3, 585 and others: write “auris” in low case first letter, no matter if the whole title is in capital first letters.

Lines 20, 21: change “comprehensive coverage of these information in a single investigation yet to be available” to “Complete coverage of this information in a single investigation is not yet available.”

Lines 34-39, a reference (recent) is required

Line 48: change “to development” to “to the development”

Line 53: “had identified” to “have been identified”

Line 61: change “others” to “other”

Line 62: change “have widely investigated” to “have been widely investigated”

Line 72: change “speculates” to “speculate”

Line 75: change “they freely live“ to “cells live freely”

Line 80: change “yet to be established” to “need to be established yet”

Line 81: change to “clinical breakpoints” to “the use of clinical breakpoints”

Line 82: change “guideline” to “guidelines”

Line 83: use low case first letter for “amphotericin”

Line 85: change “resistance” to “resistant”

Line 85: change “for resistance to fluconazole” to “for fluconazole”

Line 88: change “Thus, echinocandins” to “Echinocandins”

Line 92: you say “presented 93% of resistance for fluconazole, 35% for Amp B, and 7% for echinocandins” to “presented 93, 35 and 7% of fluconazole, Amp B and echinocandins resistance, respectively”.

Line 94, change “for isolates” to “in isolates”

Line 95: change “resistance to fluconazole (100%)” to “100% resistance to fluconazole.

Line 96: change “antifungal” to “antifungal drug”

Line 97, 127 and others: change “US” to “USA”

Line 99: change “Besides prevalence” to “Besides the prevalence analyses” Query What do you mean with “prevalence”?

Line 108: change “of cell wall” to “of the cell wall”

Lines 108, 109: change “As like C. albicans and C. glabrata” to “As occurs with C. albicans and C. glabrata”

Line 114: change “the exact resistance mechanisms were not yet elucidated” to “a probable mechanism of action has not been yet elucidated ”. The word “exact” is not adequate

Line 116: change “to loss” to “to a loss”

Line 117: add a dash: “cholesta-type”

Line 118: change “bearing” to “bear”

Line 120: change “Current approved/considered antifungal therapy.” to “Current approved/considered antifungal therapy for C. auris”

Line 125, you write “rezafungin” here and in other parts of the text. However, in line 122, you write “Rezafungin” with capital first letter. Please homogenize throughout the text.

Line 129: “through realtime clinical data” realtime?

Lines 129, 130: change ”Considering the cost, its more expensive than other antifungals, which could pose an access barrier in most of the clinical setting.” Please write in a simpler way.

For example, you can write: “The high cost of this antifungal puts a barrier to its use in clinical practice.”

Line 134: eliminate “occurrence of”

Line 135: change “molecules” to “molecule”

Line 137: was focused by the researchers??.

Line 174; change “as an emerging weapon” to “as emerging weapons”

Table 1: First line, first column: do not write “Novel Tetrazoles Featuring a Pyrazole Moiety” in capital first letter.

In the title of the first coumn you put MIC/MIC range (µg/ml). However, in the lines of below you give the MICs in µM.

In Manogepix, you put “0.004 µg/mL and 0.008”. What are the units??

Line 137: just in this line you define the abbreviation IBX for Ibrexafungerp. However, the abbrevistions must be defined in the first mentioning (line 122). In line 137 you must put only IBX.

Line 138: change “β-d-glucan” to “β-D-glucan”

Lines 140-141: a verb is lacking in the following sentence: “echinocandins which limited cross-resistance”. In addition, a comma is always required before “which”. Or you wanted to say “with”?

Line 142: after “infections”, a reference is needed.

Line 142: remined??

Line 144: change “expressed” to “showed to possess a”

Line 165: change “Another novel antifungal agent is T-2307 that belongs to the triterpenoid class” to “Another novel antifungal agent is the triterpenoid T-2307.”

Line 166: eliminate “like”

Line 168: change “with” to “to the”

Line 197: you say: “Lohse and their colleague” but the reference is not provided.

Line 269: write carvacrol with low-case first letter.

Line 279: “S. terebinthifolia” Write the full binomial name

Lines 281-295: there are non-required capital first letters, lack of spaces between the number and the unit and other errors.

Table 3, 3rd column. There is a mixture of units  and thebtitle says µg/ml. Please homogenize

Line 546-553 are not conclusions. Shift to another sections. The rest of the Conclusions section is poor. I recommend the reading of the same section in Auditeau, E., Chassagne, F., Bourdy, G., Bounlu, M., Jost, J., Luna, J., ... & Boumediene, F. (2019). Herbal medicine for epilepsy seizures in Asia, Africa and Latin America: A systematic review. J. Ethnopharmacol. 234, 119-153.

The authors must clearly revise the text. Probably there are many other errors than those found by this Editor. The conceptual errors must be corrected.

please see above

Author Response

Reviewer 2

The Review covers the different approaches previously published for the treatment of Candida auris. They include existing antifungal drugs, chemical compounds, essential oils, natural products, antifungal peptides, immunotherapy, antimicrobial photodynamic therapy, drug repurposing and drug delivery systems.

However major revision is required. They are detailed below

The manuscript is comprehensibly written

However, some major and minor corrections are required:

Major concerns

Line 233: In this phrase there is a serious conceptual error: You say: “Essential oils are the fatty acids moieties derived from bioactive plants.” Please consult to a Natural products Chemist. Essential oils are mixtures of volatile compounds that are the contrary of fatty acids, which are fixed oils. This is an error that cannot be published.

Lines 332-335: You say “These chemical substances alone or in combination with other drugs produce significant biological activities among the different targets.” Note that essential oils are not “chemical substances” but mixtures of volatile compounds, obtained by steam distillation

Answer: Thank you for your insightful comments. We appreciate the opportunity to clarify and correct these points. We apologize for this oversight and the appropriate revision is included in the revised manuscript to accurately reflect the nature of essential oils.

Essential oils are mixtures of volatile (hydrophobic liquid) compounds obtained through the steam distillation of various plant parts, such as flowers, bark, leaves, and other plant materials. Due to their unique properties, essential oils have a wide range of applications, with predominant use in aromatherapy.

Minor remarks

Lines 3, 585 and others: write “auris” in low case first letter, no matter if the whole title is in capital first letters.

Lines 20, 21: change “comprehensive coverage of these information in a single investigation yet to be available” to “Complete coverage of this information in a single investigation is not yet available.”

Line 48: change “to development” to “to the development”

Line 53: “had identified” to “have been identified”

Line 61: change “others” to “other”

Line 62: change “have widely investigated” to “have been widely investigated”

Line 72: change “speculates” to “speculate”

Line 75: change “they freely live“ to “cells live freely”

Line 80: change “yet to be established” to “need to be established yet”

Line 81: change to “clinical breakpoints” to “the use of clinical breakpoints”

Line 82: change “guideline” to “guidelines”

Line 83: use low case first letter for “amphotericin”

Line 85: change “resistance” to “resistant”

Line 85: change “for resistance to fluconazole” to “for fluconazole”

Line 88: change “Thus, echinocandins” to “Echinocandins”

Line 92: you say “presented 93% of resistance for fluconazole, 35% for Amp B, and 7% for echinocandins” to “presented 93, 35 and 7% of fluconazole, Amp B and echinocandins resistance, respectively”.

Line 94, change “for isolates” to “in isolates”

Line 95: change “resistance to fluconazole (100%)” to “100% resistance to fluconazole.

Line 96: change “antifungal” to “antifungal drug”

Line 97, 127 and others: change “US” to “USA”

Line 99: change “Besides prevalence” to “Besides the prevalence analyses” Query What do you mean with “prevalence”?

Line 108: change “of cell wall” to “of the cell wall”

Lines 108, 109: change “As like C. albicans and C. glabrata” to “As occurs with C. albicans and C. glabrata”

Line 114: change “the exact resistance mechanisms were not yet elucidated” to “a probable mechanism of action has not been yet elucidated ”. The word “exact” is not adequate

Line 116: change “to loss” to “to a loss”

Line 117: add a dash: “cholesta-type”

Line 118: change “bearing” to “bear”

Line 120: change “Current approved/considered antifungal therapy.” to “Current approved/considered antifungal therapy for C. auris”

Line 125, you write “rezafungin” here and in other parts of the text. However, in line 122, you write “Rezafungin” with capital first letter. Please homogenize throughout the text.

Line 129: “through realtime clinical data” realtime?

Lines 129, 130: change ”Considering the cost, its more expensive than other antifungals, which could pose an access barrier in most of the clinical setting.” Please write in a simpler way.

For example, you can write: “The high cost of this antifungal puts a barrier to its use in clinical practice.”

Line 134: eliminate “occurrence of”

Line 135: change “molecules” to “molecule”

Line 174; change “as an emerging weapon” to “as emerging weapons”

Table 1: First line, first column: do not write “Novel Tetrazoles Featuring a Pyrazole Moiety” in capital first letter.

Line 137: just in this line you define the abbreviation IBX for Ibrexafungerp. However, the abbrevistions must be defined in the first mentioning (line 122). In line 137 you must put only IBX.

Line 138: change “β-d-glucan” to “β-D-glucan”

Lines 140-141: a verb is lacking in the following sentence: “echinocandins which limited cross-resistance”. In addition, a comma is always required before “which”. Or you wanted to say “with”?

Line 142: after “infections”, a reference is needed.

Line 142: remined??

Line 144: change “expressed” to “showed to possess a”

Line 165: change “Another novel antifungal agent is T-2307 that belongs to the triterpenoid class” to “Another novel antifungal agent is the triterpenoid T-2307.”

Line 166: eliminate “like”

Line 168: change “with” to “to the”

Line 269: write carvacrol with low-case first letter.

Line 279: “S. terebinthifolia” Write the full binomial name

Lines 281-295: there are non-required capital first letters, lack of spaces between the number and the unit and other errors.

Table 3, 3rd column. There is a mixture of units  and thebtitle says µg/ml. Please homogenize

Line 546-553 are not conclusions. Shift to another sections. The rest of the Conclusions section is poor. I recommend the reading of the same section in Auditeau, E., Chassagne, F., Bourdy, G., Bounlu, M., Jost, J., Luna, J., ... & Boumediene, F. (2019). Herbal medicine for epilepsy seizures in Asia, Africa and Latin America: A systematic review. J. Ethnopharmacol. 234, 119-153.

Answer: Thank you very much for the detailed report on the errors in the manuscript. As suggested by the reviewer, all the above-mentioned points and other errors in the manuscript have been revised and corrected accordingly.

The authors must clearly revise the text. Probably there are many other errors than those found by this Editor. The conceptual errors must be corrected.

Answer: Thank you for pointing out this error. We have carefully reviewed the manuscript and corrected all instances.

  1. Lines 34-39, a reference (recent) is required

Answer: As per reviewer suggestion the following references are included in the revised manuscript.

  • Tamo, S. B. (2020). Candida infections: clinical features, diagnosis and treatment. Infect. Dis. Clin. Microbiol, 2, 91-103.
  • Talapko, J., Juzbašić, M., Matijević, T., Pustijanac, E., Bekić, S., Kotris, I., & Škrlec, I. (2021). Candida albicans—the virulence factors and clinical manifestations of infection. Journal of Fungi7(2), 79.
  • Pappas, P. G., Lionakis, M. S., Arendrup, M. C., Ostrosky-Zeichner, L., & Kullberg, B. J. (2018). Invasive candidiasis. Nature Reviews Disease Primers, 4(1), 1-20.
  • Lamoth, F., Lockhart, S. R., Berkow, E. L., & Calandra, T. (2018). Changes in the epidemiological landscape of invasive candidiasis. Journal of Antimicrobial Chemotherapy, 73(suppl_1), i4-i13.
  1. Line 137: was focused by the researchers??.

Answer: Yes, this information presented here was obtained from the recent publication by Wang et al., 2024. However, the entire infographic was generated independently by our team without involving any copyright issues.

  1. In the title of the first coumn you put MIC/MIC range (µg/ml). However, in the lines of below you give the MICs in µM. In Manogepix, you put “0.004 µg/mL and 0.008”. What are the units??

Answer: The appropriate modifications are included in the revised manuscript.

  1. Line 197: you say: “Lohse and their colleague” but the reference is not provided.

Answer: As per reviewer suggestion an appropriate reference is included.

Round 2

Reviewer 2 Report

The Review has been greatly improved and now is suitable for publication in Journal of Fungi.

However, when the authors receive the galley proof, please correct the following errors:

Line 17: change “specie” to “sp.” (Please note that the word “specie” must be wrriten always with “s”: “species”. However , it is better to write “sp.”

Line 25: what is “chemical medication”?: Change to a more appropriate expression.

It is Ok